# Gas chromatography–mass spectrometry analysis of fatty acids in healthy and *Aspergillus niger* MH078571.1-infected Arabica coffee beans

Amira H. Alabdalall[1,2]*

1 Department of Biology, College of Science, Imam Abdulrahman Bin Faisal University, Dammam, Saudi Arabia, 2 Basic and Applied Scientific Research Center (BASRC), Imam Abdulrahman Bin Faisal University, Dammam, Saudi Arabia

* aalabdalall@iau.edu.sa

## Abstract

The organic composition of Arabica coffee beans, particularly fatty acids, significantly influences their overall quality. After measuring its composition of fatty acids, it contained a high percentage of saturated fatty acids (SFA), including caprylic, lauric, myristic, palmitic, margaric, fat, and orchid. Moreover, the sample contained unsaturated fatty acids (USFA), namely palmitoleic acid (C16:1), oleic acid (C18:1), linoleic acid (C18:2), and alpha-linoleic acid (C18:3). Coffee beans are susceptible to infection by fungi during storage, the development of which has adverse effects on the beans. The present study aimed to examine the impact of *Aspergillus niger* MH078571.1 infection on the diversity and abundance of fatty acids in green Arabica coffee beans. The impact of *Aspergillus niger* on the consumption of fatty acids in Arabica coffee beans was assessed. The findings of the study indicate that the duration of storage had a significant impact on the levels of fatty acids, specifically miristic (C14:0), margaric (C17:0), and stearic (C18:0), which increased as the storage period and temperature increased. Conversely, the percentage of oleic acid decreased under these conditions. This trend was observed across different storage temperatures (0, 8, and 25˚C) in untreated coffee beans affected by fungal activity.

## Introduction

Historically, coffee has been known since the beginning of the ninth century B.C. (1). Coffee is consumed worldwide; most people consume coffee daily [1–3]. Some estimates say it's the second most traded commodity after oil [4]. Globally, there are 103 coffee species, 70 of which are tropical, and only three are commercially grown. Coffea arabica, robusta, liberica [5, 6].

Arabica coffee has a different aroma, flavor, and stimulating impact than Robusta [7–9]. Arabica coffee, which makes up 64% of global coffee production, is closely monitored for its organoleptic qualities. Robusta coffee accounts for 35% of global labor, has less taste, and is bitterer than Arabica. Liberica coffee is less popular and produces less globally than Robusta

**Data Availability Statement:** All relevant data are within the paper and its Supporting Information files.

**Funding:** The author received no specific funding for this work.

**Competing interests:** The authors have declared that no competing interests exist.

coffee, which is cheaper. Economically significant species are Robusta and Arabica [5]. International commerce relies on them, thus only the finest quality is acceptable [10].

Fresh beans' chemicals affect coffee's taste [11]. Coffee bean chemical composition and quality vary greatly by variety, height, soil, daily temperature variations, and growing location [12–16]. Variable volatile and nonvolatile chemical components make coffee taste different. Coffee quality is altered by pre-, post-, and export handling. Climate change and plant disease are further coffee industry risks [17].

Analyzed coffee's fatty acid content by identifying the main constituents that affect quality. The researchers focused on saturated fatty acids such arachidic, stearic, and palmitic acids as speciality coffee quality indicators. These fatty acids provide sensory excellence. However, unsaturated fatty acids including elaidic, oleic, linoleic, and linolenic acid may contribute to coffees with lower acidity, smell, body, and taste [18].

Lipids are mostly fatty acids. Triacylglycerols—three fatty acids linked to a glycerol molecule—are prevalent. Fatty acids affect glycerol's digestion and melting point [19]. All living things have fatty acids. It is the most important part of coffee beans, and green Arabica beans are abundant in fatty acids [8, 20, 21]. The majority are triglycerides, sterols, and tocopherols [22]. The oil from green Arabica coffee beans is rich in fatty acids [8, 21]. Green coffee beans include main fatty acids like linoleic and palmitic acids and secondary acids like myristic, palmitoleic, eicosinoic, behenic, and arachidic acids, according to Martin et al. [21]. The acids have three forms. Coffee oil contains 49.4–59.2% saturated fatty acids. The range of monounsaturated fatty acids is 30–9.69%, whereas polyunsaturated is 29.5–39.2% [23, 24].

Coffee oil is an emollient, skin treatment, and vital dietary component owing to its role in prostaglandin formation and cell regeneration. Skin problems are associated to coffee oil deficiency [24]. Coffee oil contains a lot of palmitic acid, a skin-protecting ingredient. Green coffee beans are rich in linoleic and palmitic acids [8].

Coffee quality depends on fatty acids. It depends on height, soil, and coffee plant location. Thus, acid comparison is used to classify coffee kinds and origins, and fatty acid concentrations are determined by these components [21, 25]. Many studies have examined how these variables affect fatty acid amount and type [12, 26–30]. Few research have examined how fungal molds affect fatty acids.

Coffee beans are fermented in 10 cm layers and sun-dried for 10–25 days. Cellulose and pectin lysis ferment 754 bacteria isolates, affecting coffee quality and shelf life [31]. lower humidity than 14.5% to prevent mold formation during storage [32].

Despite the recognized significance of fatty acids in contributing to the flavor and aroma of coffee, there is a lack of research investigating the impact of fungal molds on the composition and characteristics of fatty acids in coffee beans. As a result, this study aimed to determine how storage duration, temperature, and fungal infection affected fatty acid levels in the local market's most popular green coffee beans.

## Materials and methods

### Sample collection

Green coffee bean samples of the four major coffee bean varieties, Harari, Barry, lukkmaty, and habbashy, were obtained from local marketplaces in Dammam and Al-Khobar, in the Eastern Province of the Kingdom of Saudi Arabia.

### Preparing coffee beans for fungus infection

The study used healthy coffee beans, which were chosen from four different varieties. These beans were carefully selected to ensure that they did not exhibit any defects. Each variety was then divided into two groups: one group served as the control, while the other group was

subjected to fungus infection. To ensure sterilization, Arnold's sterilizer was employed. The calculation of relative humidity was performed. After sterilization, the humidity was adjusted to 10% for all samples, both the control group and the other group that was industrially contaminated with *Aspergillus niger* MH078571.1, and this strain was previously identified as the highest lipase-producing *Aspergillus niger* MH078571.1strain [33]. *Aspergillus niger* MH078571.1 was cultured on Potato Dextrose Agar (PDA) medium for 15 days at 25 2˚C. Spores from pure culture suspended in sterilized distilled water containing 0.1% agar were used as an inoculum. The spore density was determined by using a hemacytometer to count 10 samples of fungal suspension. The calculated quantity of water required to raise the moisture content of the seed to 10% was supplemented with a spore suspension yielding a final concentration of approximately 2000–2500 spores/g. Three replicates were prepared for each treatment and incubated at 0, 8, and 25˚C for 3 and 9 months, then fatty acids were measured.

### Bean fatty acid estimation

Brockerhoff [34] using GLC to identify fatty acids in coffee bean hexane extracts:

### Fatty acid extraction/methylation

The coffee oil obtained through hexane extraction. The solvent was subjected to reduced pressure, resulting in its evaporation. The remaining substance was then subjected to saponification in a boiling water bath for a duration of 30 minutes. This process involved the use of an alcoholic solution of sodium hydroxide with a concentration of 15% w/v. Following the cooling process, a volume of 20 mL was subsequently transferred into a separatory funnel. The mixture underwent vigorous agitation subsequent to the addition of petroleum ether (60–80˚C). The aqueous layer was acidified using a 1:1 ratio of hydrochloric acid (HCl). The free fatty acids were subjected to a triple extraction process using petroleum ether (60–80˚C) in 10 mL aliquots. The solvents were subsequently recycled by rinsing them with distilled water until the pH indicator phenolphthalein reached a neutral state. Following the removal of moisture using anhydrous sodium sulfate, the petroleum ether extracts were subjected to evaporation under reduced pressure. Dissolving fatty acids (1–2 mg) in anhydrous diethyl ether (0.5:1 mL), we then added diazomethane solution drop by drop until the yellow hue was maintained and the nitrogen gas bubbles ended. In 15 minutes at room temperature, the solvent was gone. The last step included dissolving fatty acid methyl esters in chloroform so that they could be detected in aliquots using gas-liquid chromatography.

### Gas-liquid chromatography of fatty acid methyl esters

For the gas chromatography with the flame ionization detector and the 1.5 by 4 mm coiled glass column with the 10% PEGA supported by the acid washed diatomic C (100–120 mesh), the GCV pye Unicam was used. The sample was injected into the column using Hamilton micro syringes. Column temperature in isothermal gas chromatography is 190˚C, detector temperature is 300˚C, and injection temperature is 250˚C. Airflow of 330 mL min-1, hydrogen flow of 33 mL min-1, and nitrogen flow of 30 mL min-1. Average chart speed is 1 cm every 2 minutes, with a precision of 32 x $10^{-2}$.

### Identification and quantification

Accurate materials, such as methyl esters of 10:0–20:0 fatty acids, were employed to determine the retention times of new fatty acids. The Philips PU 4810 computer integrator was used for peak detection and quantification.

## Statistical analysis

Using SPSS version 23 [35], we performed an analysis of variance (ANOVA) to look for correlations between the control parameters and fungal infection. The data obtained were expressed as mean + standard deviation (SD) of three replicates.

## Results

The composition of coffee oil in all four varieties examined consisted of both saturated and unsaturated fatty acids at varying levels, which were specific to each coffee variety. The saturated fatty acids present included Caprilic acid (C10:0), Laurie acid (C12:0), Miristic acid (C14:0), Palmitic acid (C16:0), Margaric acid (C17:0), Stearic acid (C18:0), Arachidic acid (C20:0), and Eicosenoic acid (C22:0). Additionally, the unsaturated fatty acids detected were Palmitoleic acid (C16:1), Oleic acid (C18:1), Linoleic acid (C18:2), and Alpha-Linoleic acid (C18:3).

The present study investigated the impact of fungal infection and storage conditions on the composition of fatty acids in coffee beans. Our findings indicate that both control and treated beans exhibited the presence of saturated and unsaturated fatty acid groups. The data presented in S1-S8 Tables in S1 File indicate that miristic, margaric, stearic, oleic, and alpha-linoleic acids exhibited the highest frequencies, while the remaining types displayed negligible proportions (3%). The fatty acid composition was found to be affected by the duration of storage or the presence of fungal contamination. Fluctuations in temperatures were found to induce alterations in acid percentages. The findings indicated that the constituents of the bean were influenced by varying durations of storage. The fatty acid composition, specifically the presence of Oleic acid, in control samples of Harari, Barry, Lukkmaty, and Habashy beans, was significantly influenced by the storage temperature. After a period of 3 months at a temperature of 25˚C, the respective percentages of Oleic acid were recorded as 42.97%, 41.67%, 42.56%, and 41.86%. Subsequently, after a duration of 9 months, the percentages of Oleic acid for the same bean varieties were observed as 13.76%, 24.26%, 19.76%, and 27.02%. The degradation of the samples infected with *Aspergillus niger* was observed to intensify as the percentages of Oleic fatty acids decreased. Specifically, after a period of 3 months, the percentages of Oleic fatty acids reached values of 40.29%, 39.53%, 41.76%, and 38.65% respectively under the same conditions. Furthermore, a significant reduction in oleic fatty acid content was observed in all tested varieties after 9 months, with percentages recorded as 10.15%, 17.02%, 18.79%, and 22.24% respectively.

The impact of storage duration on the fatty acid composition was observed in the four tested varieties of green coffee beans, namely Harari, Barry, Lukkmaty, and Habbashy. These varieties exhibited significant levels of saturated and unsaturated fatty acids, which were found to be influenced by both the storage period and temperature. The findings of the study indicate that the Habbashy variety exhibited a higher proportion of saturated acids, specifically 82.65%, when subjected to a temperature of 0˚C for a period of 9 months. In comparison, the Lukkmaty variety displayed a slightly lower percentage of saturated acids at 81.54% under identical conditions. At a temperature of 8˚C, the fatty acid compositions for the Barry and Harari cultivars were found to be 77.74% and 77.12%, respectively, as indicated in Table 1.

The results in Table 1 and Fig 1 show that the four studied varieties contained saturated and unsaturated fatty acids in different proportions, and it was noted that the total unsaturated fatty acids decreased to less than half in samples stored for 9 months compared to 3 months in all varieties.

In contrast, the saturated acids increased, and the results showed that this decrease and increase were greater in samples treated with *Aspergillus niger* (Fig 2).

**Table 1. Total saturated and unsaturated fatty acids of the four types of green coffee beans after 3 and 9 months of artificial infection with *Aspergillus niger*.**

| Fatty acids | Varaity/period | control (100%) | | | *A. niger* (100%) | | |
|---|---|---|---|---|---|---|---|
| | | 0°C | 8°C | 25°C | 0°C | 8°C | 25°C |
| SFA | Harari/ 3 months | 52.51 | 53.45 | 52.53 | 55.9 | 56.56 | 55.09 |
| USFA | | 46.68 | 43.87 | 46.28 | 44.09 | 43.46 | 43.77 |
| TFA | | 99.19 | 97.32 | 98.81 | 99.99 | 98.92 | 98.86 |
| SFA | Harari/ 9 months | 77.34 | 63.53 | 73.98 | 70.2 | 70.51 | 76.34 |
| USFA | | 20.76 | 36.44 | 25.56 | 22.65 | 22.19 | 22.82 |
| TFA | | 98.1 | 99.97 | 99.54 | 92.85 | 92.7 | 99.16 |
| SFA | Barry / 3 months | 54.05 | 54.1 | 64.61 | 56.23 | 53.59 | 60.3 |
| USFA | | 45.9 | 45.83 | 35.32 | 43.53 | 46.31 | 39.6 |
| TFA | | 99.95 | 99.93 | 99.93 | 99.76 | 99.9 | 99.9 |
| SFA | Barry / 9 months | 71.61 | 77.12 | 70 | 78.23 | 60.64 | 71.94 |
| USFA | | 28.32 | 22.83 | 29.95 | 21.42 | 39.3 | 27.64 |
| TFA | | 99.93 | 99.95 | 99.95 | 99.65 | 99.94 | 99.58 |
| SFA | Lukkmaty / 3 months | 53.69 | 58.09 | 54.46 | 54.52 | 64.7 | 63.32 |
| USFA | | 46.25 | 41.95 | 45.5 | 45.32 | 35.21 | 36.57 |
| TFA | | 99.94 | 99.96 | 99.96 | 99.84 | 99.91 | 99.89 |
| SFA | Lukkmaty / 9 months | 76.53 | 79.08 | 81.54 | 77.42 | 79.86 | 80.97 |
| USFA | | 23.41 | 21.68 | 18.41 | 22.7 | 20.02 | 18.95 |
| TFA | | 99.94 | 99.98 | 99.95 | 99.62 | 99.88 | 99.92 |
| SFA | Habbashy / 3 months | 54.41 | 55.56 | 55.7 | 57.52 | 57.27 | 53.92 |
| USFA | | 45.55 | 44.37 | 44.25 | 42.32 | 42.69 | 46.06 |
| TFA | | 99.96 | 99.93 | 99.95 | 99.84 | 99.96 | 99.98 |
| SFA | Habbashy / 9 months | 70.34 | 75.07 | 82.65 | 73.86 | 75.4 | 80.58 |
| USFA | | 29.63 | 24.76 | 17.32 | 26.09 | 23.47 | 15.66 |
| TFA | | 99.97 | 99.83 | 99.97 | 99.95 | 98.87 | 96.24 |

Saturated fatty acids (SFA), unsaturated fatty acids (USFA), Total fatty acids (TFA).

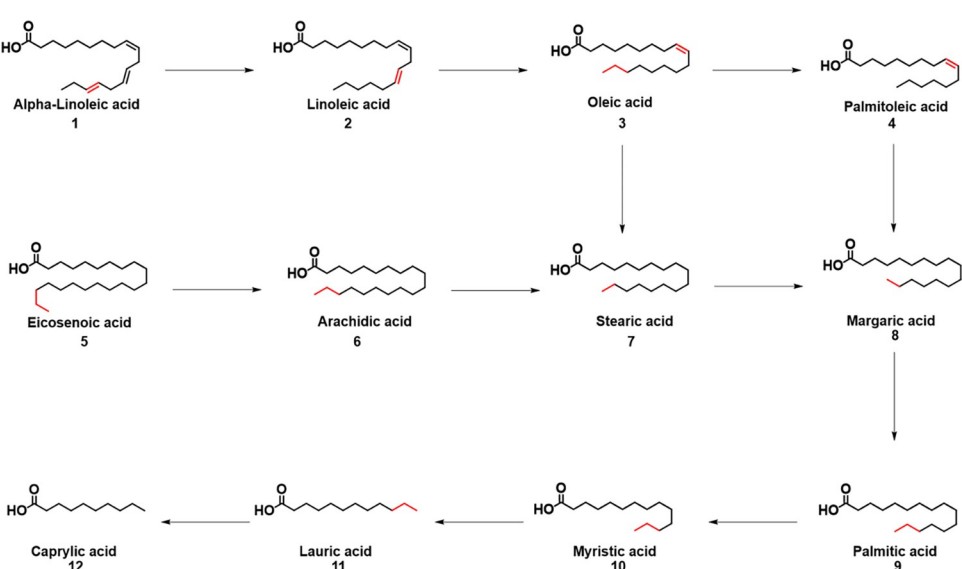

**Fig 1. The proposed mechanism of fatty acid breakdown involves hydrogenation, which converts unsaturated fatty acid into saturated fatty acid.**

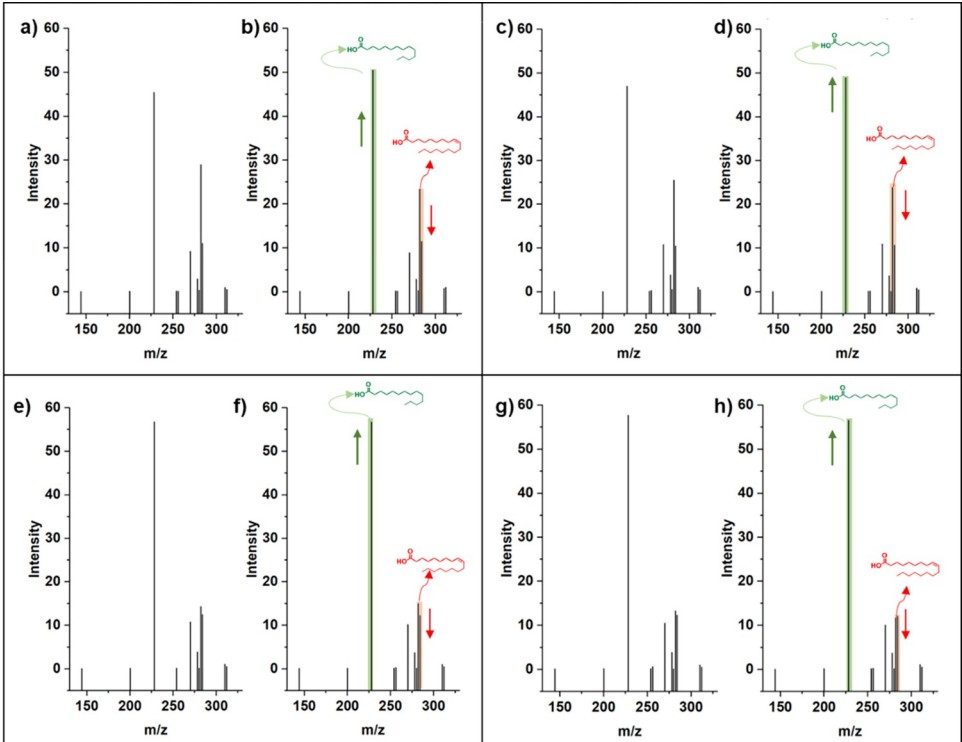

**Fig 2. Displays the chromatogram depicting the composition of fatty acids found in a sample of Arabica coffee beans.** a) Harari control, b) Harari *A. niger*, c) Barry control, d) Barry *A. niger*, e) Lukkmaty control, f) Lukkmaty *A. niger*, g) Habbashy control, and h) Habbashy *A. niger*.

Artificial fungal inoculation of beans revealed fungal infection of beans destroyed fatty acids, especially unsaturated ones. After 9 months at a temperature of 25˚C, it decreased to 22.82, 27.64, 18.95, and 15.66% for Harari, Barry, Lukkmaty, and Habbashy cultivars, respectively (Table 1).

## Discussion

The findings of the study indicated that the concentrations of saturated fatty acids (SFA), including stearic, margaric, and meristic acid, exhibited an upward trend following infection and prolonged storage durations, and elevated temperatures across all varieties of coffee beans. The levels of these fatty acids varied depending on the specific coffee variety being studied. The green coffee bean was found to contain two predominant acids, namely miristic acid and oleic acid [36–40]. In a similar vein, it was observed that unsaturated fatty acids (USFA) exhibited a reduction in quantity because of both infection and storage, akin to oleic acid. These phenomena could potentially arise from the transfer of fatty acids to other molecules or from one variety to another. This finding was consistent with the findings of various researchers [36–39, 41, 42].

The presence of *A. niger* infection in beans resulted in a greater reduction of oleic fatty acid compared to the control samples across all treated groups. This decrease was observed to be more pronounced as the storage period increased, likely due to the inherent propensity of oleic fatty acid to undergo reactions and convert into other fatty acids [43, 44]. Furthermore, it was observed that the concentrations of miristic, margaric, and stearic acids, as well as fatty acids, exhibited an upward trend in infected beans as the storage period progressed. This

phenomenon can potentially be attributed to the heightened hydrolytic activity of certain enzymes, such as lipase, and the subsequent degradation of fatty acids [44]. Additionally, it was noted that the functionality of lipase enzymes in sunflowers is influenced by both the temperature and duration of storage. In their study, Ghasemnez et al. [43] found similar outcomes for evening primrose as those reported by [45] when examining soybeans. Research has revealed that unsaturated fatty acids, including oleic, palmetiolic, linoleic, and alpha-linoleic acids, undergo rapid oxidation when exposed to oxygen. Consequently, an extended duration of storage is associated with a corresponding elevation in the rate of oxidation, and conversely. According to [46], a number of enzymes that play a role in the metabolic process of fatty acids, including acyl oxidase, malate synthase, and citrate synthase, are dependent on the presence of oxygen for their proper functioning. According to [44], variations in temperature have distinct impacts on the levels of fatty acids found in coffee beans.

The experimental conditions employed in this study promote the proliferation of *A. niger* and its enzymatic functions, leading to the degradation of seed tissues and subsequent deterioration. These findings align with previous research conducted by [47–49], which demonstrated a direct correlation between the decomposition of fava beans and factors such as humidity, temperature, and duration of storage. Additionally, the study conducted by [50, 51] observed an augmentation in tissue pigmentation in maize grains when exposed to mildews. In their seminal work, Cole and Milner [52] expounded upon the phenomenon of bean coloration, attributing it to a chemical reaction involving nitrogenous compounds and reducing sugars. According to [53], the presence of brown coloration in the inner tissues can be attributed to the process of fatty acid oxidation. The quality of coffee and its flavor components have been attributed to the composition of fatty acids present in the coffee, as stated by [18] in their 2015 study. In a study conducted by [54], it was observed that bacteria and fungi are capable of synthesizing multiple enzymes involved in cell wall metabolism. These enzymes are responsible for cleaving the methyl ester bonds found in ferulic and coumaric acids, thereby enhancing the activity of hemicellulose enzymes during the hydrolysis process of cinnamic acid [55].

Several studies have highlighted the impact of storage duration, humidity, and temperature on product quality, including the works of [54, 55]. Several studies have indicated that the oil content in sunflowers tends to decrease after a storage period of three months. The decrease in oil content can be attributed to the combined effects of elevated temperature and humidity during storage, which promote oil decomposition and lead to an increase in free fatty acids. Additionally, this phenomenon has been reported by several previous studies [56–58]. In a study conducted by [59], the fatty acid levels of two coffee samples were examined. These samples were vacuum-packed and stored at a temperature of 25˚C for a duration of 180 days. The first sample consisted of pure Arabica beans, while the second sample was a blend of 80% Arabica and 20% Robusta beans. The findings of the study revealed that only the second sample exhibited elevated levels of fatty acids. In their study, Ghasemnez and Honermeier [44] documented a reduction in the levels of oleic acid from 90.6% to 88.2% over a period of 4 months during storage. They attributed this decline to the acid's propensity to undergo reactions and transition into free forms. A parallel effect of palmalic and linoleic acids has been identified. In their study, Ghasemnez and Honermeier [44] observed a reduction in the concentration of oleic acid from 90.6% to 88.2% following a storage period of 4 months. The authors attributed this decline to the activity of enzymes, specifically lipase. The explanation provided elucidates the acid's capacity to engage in interactions and transitions towards liberated states, a phenomenon that is similarly observed in palmalic and linoleic acids.

Moon et al., [40] investigated the causes of coffee losing its quality and distinct flavor. They confirmed that the oxidation processes that occur to fatty acids for example, when kept at high temperatures and for more extended storage periods, lead to increased oxidation processes in

fats and a loss of taste quality. The first stage in tasting coffee is Olfaction since the aroma is perceived before the flavor. Coffee odor is attributed to volatile compounds in the brew activating neurons in the nasal canal. Caffeine, trigonelline, chlorogenic acid, and volatile compounds such as terpenoids are all components of a well-balanced cup of coffee [60].

According to the findings of Shahidi and Hossain [61], the process of oxidation of unsaturated fatty acids leads to the formation of hydroperoxides, which subsequently undergo decomposition to produce volatile secondary lipid oxidation products such as aldehydes, alcohols, and ketones, which are known to contribute to the development of odor.

Unsaturated fatty acids (USFA), including Palmitic, Stearic, and Alpha-Linoleic acids, were found in higher concentrations than saturated fatty acids (SFA). During storage, the amount of USFA dropped while the quantity of SFA increased.

Longer periods of storage resulted in a decrease in Stearic, Oleic, and Linoleic acid due to oxidation, whereas shorter periods of storage increased Palmitic and Arachidic acid. Acidity, aroma, body, and flavor all become less evident with greater USFA levels. Therefore, the high level of SFA in specialty coffees, especially palmitic, stearic, and arachidic acids [18, 62], contribute to their distinctive sensory character. However, the drink takes on a disagreeable taste and odor owing to the high quantities of polyunsaturated fatty acids, which is caused by the simple breakdown of double bonds and the creation of foul-flavored chemicals [63].

## Conclusion

From the data obtained from this study, it has been concluded that all of the cultivars' grains were found to have various concentrations of 11 distinct saturated and unsaturated fatty acids. At the conclusion of the storage time, oleic acid was found to have decreased drastically, whereas Miristic, margaric, and stearic acids had increased; further degradation was seen as a result of fungal infection.

## Supporting information

**S1 File. GC-MS analysis of fatty acids in healthy and *Aspergillus niger* MH078571.1-infected Arabica coffee beans.**
(DOCX)

## Acknowledgments

The authors would like to acknowledge Dr. Fatimah A. Aldakheel- Polymer Synthesis Laboratory, KAUST Catalysis Center, Physical Sciences and Engineering Division, King Abdullah University of Science and Technology (KAUST) for sharing her knowledge and experience.

## Author Contributions

**Conceptualization:** Amira H. Alabdalall.

**Data curation:** Amira H. Alabdalall.

**Formal analysis:** Amira H. Alabdalall.

**Investigation:** Amira H. Alabdalall.

**Methodology:** Amira H. Alabdalall.

**Project administration:** Amira H. Alabdalall.

**Resources:** Amira H. Alabdalall.

**Software:** Amira H. Alabdalall.

**Supervision:** Amira H. Alabdalall.

**Validation:** Amira H. Alabdalall.

**Visualization:** Amira H. Alabdalall.

**Writing – original draft:** Amira H. Alabdalall.

**Writing – review & editing:** Amira H. Alabdalall.

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
