## [Decision Letter · Decision Letter 0]

11 Sep 2023

PONE-D-23-23915GC-MS analysis of fatty acids in healthy and Aspergillus niger MH078571.1-infected Arabica coffee beans.PLOS ONE

Dear Dr. Alabdalall

Thank you for submitting your manuscript to PLOS ONE. After careful consideration, we feel that it has merit but does not fully meet PLOS ONE’s publication criteria as it currently stands. Therefore, we invite you to submit a revised version of the manuscript that addresses the points raised during the review process.

We look forward to receiving your revised manuscript.

Kind regards,

Awatif Abid Al-Judaibi, PhD

Academic Editor

PLOS ONE

Reviewers' comments:

Reviewer's Responses to Questions

**Comments to the Author**

1. Is the manuscript technically sound, and do the data support the conclusions?

Reviewer #1: Yes

Reviewer #2: Partly

Reviewer #3: Yes

2. Has the statistical analysis been performed appropriately and rigorously? 

Reviewer #1: Yes

Reviewer #2: Yes

Reviewer #3: No

3. Have the authors made all data underlying the findings in their manuscript fully available?

Reviewer #1: Yes

Reviewer #2: Yes

Reviewer #3: Yes

4. Is the manuscript presented in an intelligible fashion and written in standard English?

Reviewer #1: Yes

Reviewer #2: Yes

Reviewer #3: No

5. Review Comments to the Author

Reviewer #1: • This is an intriguing research that sought to investigate the effect of Aspergillus niger infection on the variety and abundance of fatty acids in green Arabica coffee beans, as well as the effect of Aspergillus niger on the intake of fatty acids in Arabica coffee beans.

• Title: It is not preferable to use abbreviations in the title for more clarity.

• Abstract: The abstract offers an accurate summary of the paper, and the language used in the abstract is easy to read and understand.

• Abbreviations should be defined at first mention and used consistently thereafter such as Aspergillus niger (A. niger, saturated fatty acids (SFA), unsaturated fatty acids (USFA),...

• Please rearrange key words alphabetically

Introduction:

• Introduction: Too long, it should cover three parts: the background, the scientific significance, and the aim of the study. It is noted that there are repeated paragraphs or cover the same idea, so please shorten the introduction to include these parts for more clarity.

• Correction: Mekete [1] indicated that coffee has a history; it has been known since the beginning of the ninth century B.C. Coffee. Change as: Historically, coffee has been known since the beginning of the ninth century B.C. (1).

• Correction: Coffee beans are fermented in 10 cm thick layers and dried in the sun for 10-25 days. Approximately 754 microbial isolates are fermented by cellulose and pectin lysis; these fermentations impact coffee quality and shelf life [31]. Change as: It has been found that fermentations impact coffee quality and shelf life through fermentation of coffee beans in 10 cm thick layers and drying in the sun for 10-25 days and approximately fermentation of 754 microbial isolates by cellulose and pectin lysis (31).

Materials and Methods:

• Fatty acid extraction/methylation: It is preferable to add a figure representing the extraction of fatty acids by gas chromatography, showing the conditions used with the adopted reference for more clarity.

Results:

• The findings are presented clearly, and the results are reliable.

• Table 1: Clear and comprehensive. However, it is preferable to clarify the abbreviations in the lower margin of the table for more clarity.

Discussion:

• Please indicate the reference number immediately after the researchers' names, for example: Ulla et al. (45) and so on.

• Correction: These findings align with previous research conducted by [47-49], which demonstrated a direct...Change as: These findings align with previous research [47-49], which demonstrated a direct...

• Correction: Several studies have highlighted the impact of storage duration, humidity, and temperature on product quality, including the works of [54, 55]...Change as: Several studies have highlighted the impact of storage duration, humidity, and temperature on product quality, including previous works [54, 55].

• Correction: [56, 57, 58] have all reported this phenomenon. Change as: Additionally, this phenomenon has been reported by several previous studies [56, 57, 58].

Conclusion:

• It is preferable to focus on the conclusion of the study, so it is preferable to rephrase it as follows:

From the data obtained from this study, it has been concluded that all of the cultivars' grains were found to have various concentrations of 11 distinct saturated and unsaturated fatty acids. At the conclusion of the storage time, oleic acid was found to have decreased drastically, whereas Miristic, margaric, and stearic acids had increased; further degradation was seen as a result of fungal infection.

Reviewer #2: The manuscript describes the quantities of fatty acids in green coffee beans with storage at various conditions, and which incubation. The topic is interesting, but lacks some details in the methods, and some results as well. here are my comments:

-in the coffee varieties, you seem to include brands, you include the scientific name of the variety, composition, country of origin, there is no need for brands

- all the manuscript are very lengthy and not summarized making it hard to read by the reader. For example a very lengthy introduction isn't required and out of scope, no need for the history of coffee. That does not serve the study.

- The impact of fatty acids composition on the quality is kind of subjective (different cultures enjoy different coffees), moreover it has not been well-addressed. How to improve the quality? which fatty acids should be higher (again this will be affected by the region).

- in the table, the legend isn't very clear. what is the units in the table ( it is not mentioned). if it is %, why they donot add up to 100%.

- the inoculum of fungus is not mentioned in the methodology. No microbiology at all in the manuscript. I would suggest to include the colony forming units (cfu) for fungus at various conditions to verify contamination.

- the conclusion should give us an idea of whether the quality has been preserved or not.

Reviewer #3: The method is not described appropriately.

Table legends are not present

Tables need to include the results of the statistical analysis(i.e. p-value)

Statistical analyses data is not presented

Nothing in the discussion describe the effect of A. niger MH078571.1

6. PLOS authors have the option to publish the peer review history of their article (what does this mean?). If published, this will include your full peer review and any attached files.

Reviewer #1: No

Reviewer #2: No

Reviewer #3: **Yes: **Rana Abutaima

---

## [Author Response · Author response to Decision Letter 0]

21 Sep 2023

The response to the reviewers' comments has been attached as other attachments

---

## [Decision Letter · Decision Letter 1]

11 Oct 2023

Gas chromatography–mass spectrometry analysis of fatty acids in healthy and Aspergillus nigerMH078571.1-infected Arabica coffee beans.

PONE-D-23-23915R1

Dear Dr. Amira H. Alabdalall,

We’re pleased to inform you that your manuscript has been judged scientifically suitable for publication and will be formally accepted for publication once it meets all outstanding technical requirements.

Kind regards,

Awatif Abid Al-Judaibi, PhD

Academic Editor

PLOS ONE

Reviewers' comments:

Reviewer's Responses to Questions

**Comments to the Author**

1. If the authors have adequately addressed your comments raised in a previous round of review and you feel that this manuscript is now acceptable for publication, you may indicate that here to bypass the “Comments to the Author” section, enter your conflict of interest statement in the “Confidential to Editor” section, and submit your "Accept" recommendation.

Reviewer #1: All comments have been addressed

Reviewer #2: All comments have been addressed

Reviewer #3: All comments have been addressed

2. Is the manuscript technically sound, and do the data support the conclusions?

Reviewer #1: Yes

Reviewer #2: Yes

Reviewer #3: Yes

3. Has the statistical analysis been performed appropriately and rigorously? 

Reviewer #1: Yes

Reviewer #2: Yes

Reviewer #3: Yes

4. Have the authors made all data underlying the findings in their manuscript fully available?

Reviewer #1: Yes

Reviewer #2: Yes

Reviewer #3: Yes

5. Is the manuscript presented in an intelligible fashion and written in standard English?

Reviewer #1: Yes

Reviewer #2: Yes

Reviewer #3: Yes

6. Review Comments to the Author

Reviewer #1: The author has made all necessary changes and implemented all previous recommendations. In addition, addressed all my concerns.

Reviewer #2: (No Response)

Reviewer #3: (No Response)

7. PLOS authors have the option to publish the peer review history of their article (what does this mean?). If published, this will include your full peer review and any attached files.

Reviewer #1: No

Reviewer #2: No

Reviewer #3: No

---

## [Editor Report · Acceptance letter]

30 Oct 2023

PONE-D-23-23915R1 

Gas chromatography–mass spectrometry analysis of fatty acids in healthy and *Aspergillus niger* MH078571.1-infected Arabica coffee beans. 

Dear Dr. Alabdalall:

I'm pleased to inform you that your manuscript has been deemed suitable for publication in PLOS ONE. Congratulations! Your manuscript is now with our production department. 

Kind regards, 

on behalf of

Professor Awatif Abid Al-Judaibi 

Academic Editor

PLOS ONE